# Adherence to Pre-Exposure Prophylaxis (PrEP) among Men Who Have Sex with Men (MSM) in Portuguese-Speaking Countries

**DOI:** 10.3390/ijerph20064881

**Published:** 2023-03-10

**Authors:** Alvaro Francisco Lopes de Sousa, Shirley Veronica Melo Almeida Lima, Caíque Jordan Nunes Ribeiro, Anderson Reis de Sousa, Nilo Manoel Pereira Vieira Barreto, Emerson Lucas Silva Camargo, Agostinho Antônio Cruz Araújo, Allan Dantas dos Santos, Márcio Bezerra-Santos, Mariana dos Reis Fortunato, Matheus Arantes Possani, Adriano José dos Santos, João Lucas Tavares de Lima, Elucir Gir, Inês Fronteira, Isabel Amélia Costa Mendes

**Affiliations:** 1Hospital Sírio-Libânes, Instituto de Ensino e Pesquisa, São Paulo 01308-050, São Paulo, Brazil; 2Global Health and Tropical Medicine, Instituto de Higiene e Medicina Tropical, Universidade Nova de Lisboa, 1349-008 Lisboa, Portugal; 3Collective Health Research Center, Universidade Federal de Sergipe, Lagarto 49400-000, Sergipe, Brazil; 4Graduate Program in Nursing, Universidade Federal de Sergipe, São Cristóvão 49100-000, Sergipe, Brazil; 5Graduate Program in Nursing and Health of the Nursing School, Universidade Federal da Bahia, Salvador 40170-110, Bahia, Brazil; 6Ribeirão Preto College of Nursing, Universidade de São Paulo, Ribeirão Preto 14040-902, São Paulo, Brazil; 7Medical and Nursing Science Center, Universidade Federal de Alagoas, Arapiraca 57309-005, Alagoas, Brazil; 8Medical School, Universidade Federal do Triângulo Mineiro, Uberaba 38025-180, Minas Gerais, Brazil

**Keywords:** HIV, MSM, pre-exposure prophylaxis, sexual behavior, Brazil, Portugal, global health

## Abstract

Strengthening strategies to improve adherence to the use of pre-exposure prophylaxis (PrEP) in key populations constitutes a global health priority to be achieved across countries, especially in countries that share a high flow of people such as Brazil and Portugal. This study aimed to analyze the factors associated with adherence to PrEP among MSM from two Portuguese-speaking countries, highlighting the opportunities and preventive strategies for the global health scenario. This was a cross-sectional analytical online survey conducted from January 2020 to May 2021 with MSM in Brazil and Portugal. For analysis of the data, the Poisson regression model was used to estimate the prevalence ratio (PR) for developing a model to evaluate the associated factors in both countries in a comparative and isolated way. Adherence to PrEP use corresponded to 19.5% (*n* = 1682) of the overall sample: 18.3% (*n* = 970) for Brazil and 21.5% (*n* = 712) for Portugal. Having more than two sex partners in the last 30 days (aPR: 30.87) and routinely undergoing HIV tests (aPR: 26.21) increased the use of this medication. Being an immigrant (PR: 1.36) and knowing the partner’s serological status (PR: 1.28) increased adherence to PrEP in Portugal, whereas, in Brazil, it was being an immigrant (PR: 0.83) and not knowing the serological status (PR: 2.24) that promoted the use of this medication. Our findings reinforce the need to invest in programs and strategies to improve access and adherence to PrEP, especially in key populations.

## 1. Introduction

Notably, pre-exposure prophylaxis (PrEP) is one of the main advances in the fight against the dissemination of the HIV infection in vulnerable groups [1]. Usually, PrEP is a strategy that consists of taking a tablet orally every day before sexual exposure, although it also has shown efficacy and safety in alternative modalities of use such as on demand and injections [2,3,4,5,6]. To ensure the effectiveness of the strategy, PrEP users are typically monitored every three months with follow-up HIV and sexually transmitted infection (STI) test and other relevant laboratory tests to ensure the effectiveness of the medication, address any concerns or side effects, and promote the overall health and wellbeing of the person taking PrEP [3,4,5,6].

According to the Joint United Nations Program on HIV/AIDS (UNAIDS), in 2020, 130 countries reported having adopted the WHO’s recommendations on oral PrEP use in their national guidelines, and another 23 countries have planned to adopt the recommendations by 2023. Although these are encouraging numbers, the fact that nearly 97% of the global PrEP users are from only 30 countries is worrisome [7].

There are several issues that permeate the implementation and success of PrEP in a country [8], such the willingness and technological capacity of the governments to implement public policies comprehensively, through determination of the key populations, unmet needs and acceptance targets. For example, while PrEP use has increased among MSM in the United States, the acceptance rates remain sluggish, with less than 9% of those at a substantial risk of HIV infection actively taking PrEP, in addition to stark and persistent disparities by race/ethnicity and insurance status [9].

The main barriers to adherence include low awareness and knowledge about PrEP, a low perceived risk of HIV, high costs, fear of stigma, concerns about side effects and the perceived burden of taking a tablet every day, in addition to structural obstacles such as lack of access to PrEP care and language barriers for migrant populations [10,11,12,13,14,15].

In this context, strengthening strategies to improve the adherence to PrEP use in key populations constitutes a global health priority to be achieved across countries, especially those with low technological incorporation and lower levels of human development, such as access to good quality health services, education and socioeconomic conditions that are favorable for subsistence [16]. Thus, monitoring adherence among individuals included in PrEP programs is of significant importance in this clinical-epidemiological scenario, representing a major challenge for health services due to the limited methods available [17,18].

Brazil and Portugal are two countries that share historical, cultural and linguistic similarities, in addition to an intense and robust migratory flow between their populations that has intensified since 2017 and which is subjected to the specificities inherent in the health systems of each country [19,20,21,22]. Even with encouraging advances, HIV/AIDS control is still a challenge in these countries, which have an epidemic of the concentrated type, characterized by epidemiological contexts in which the infection disproportionately affects key populations, although, in the general population, it is below 1% [23,24].

In Brazil, transgender people are estimated to have an HIV prevalence rate of 30%; MSM have a rate of 18.3%; in people who inject drugs, it is 5.9%; and in prisoners, the rate is 4.5% [23], while in Portugal, the prevalence is, 14.8% for MSM, 7% for those who inject drugs and 3% for prisoners [25]. In addition, the incredible disproportionate growth of HIV in immigrants in this country stands out, accounting for 32% of all new infections registered [26,27].

Brazil is a Latin American country that invests heavily in prevention, with the main efforts being in key populations [28], although financial investment in the five pillars of primary prevention remains insufficient and PrEP, although available in the public sector, is underused in the country, with a notorious loss to follow-up of the users [29,30]. Portugal achieved important advances in HIV prevention and famously achieved all the objectives established in the United Nations program to combat HIV/AIDS, known as 90/90/90: 90% of those infected diagnosed, 90% on treatment and 90% with an undetectable viral load [31].

Implementation of PrEP free of charge through the national health systems of these two countries began in 2017, with extensive expansion in later years [18], although few studies have evaluated the success of its implementation [27,32,33,34] and no survey has focused on studying the population of both countries together, much less comparing them [32,33,34,35]. Questions still remain whether PrEP can be translated to a successful public health intervention, leading to a decrease in the population-level incidence of HIV, if it is treated only as a local policy, without countries considering the dynamics of the flows of migration and information between countries, for example [36].

At the global level, the advancement of technologies that establish the development of combined prevention strategies for protection against HIV has been facing difficult situations. In the Brazilian context, for example, there are problems such as low public funding of such actions, shortcomings in the creation and implementation of public policies and deficits in communication and health education strategies [37]. On the other hand, in Portugal, PrEP is only available in a few specialized clinics and hospitals, which limits its accessibility to people living outside of major cities. Moreover, there are complaints about the difficulties and bureaucracy involved in accessing the system, as well as the ways and places used to dispense the medication [38]. In addition, the government has recognized that the strategy covers very little of the necessary target population; however, the expansion of free PrEP in the country has shown modest growth over the years [25,26,27]. This has led to low spread and adherence to PrEP in key key populations.

PrEP is extremely effective in preventing the acquisition of HIV when taken properly as prescribed. It is an essential element in combined HIV prevention and is needed to achieve the Sustainable Development Goals (SDGs) and end the AIDS epidemic by 2030 [39]. It is with this international scenario in mind that this study aimed to analyze the factors associated with adherence to PrEP among MSM from two Portuguese-speaking countries (Brazil and Portugal), highlighting the differences, similarities, opportunities and preventive strategies for the global health scenario.

## 2. Materials and Methods

### 2.1. Type of Study

This was a cross-sectional and analytical study that was part of the “In_PrEP” project carried out in Brazil and Portugal from January 2020 to May 2021 under the leadership of the Hygiene and Tropical Medicine Institute (Instituto de Higiene e Medicina Tropical, IHMT) in partnership with Universidade de São Paulo (USP).

### 2.2. Population, Sample and Eligibility Criteria

This study included a random sample of MSM aged over 18 years old who had lived in Brazil or Portugal for at least 3 months, either natives of these countries or immigrants from any of the following nine Portuguese-speaking countries: Angola, Brazil, Cape Verde, Equatorial Guinea, Guinea-Bissau, Mozambique, Portugal, East Timor, and Saint Thomas and Prince.

A simple calculation of the proportion of the sample was performed using G*Power software (version 3.1.9.7; Heinrich Heine University Düsseldorf, Düsseldorf, Germany), considering the population of men over 18 years of age in both countries, with a presumed prevalence of 50% (aiming to maximize the sample and bearing in mind that this is a phenomenon for which there are still no data on prevalence), a tolerable standard error of 3% and a confidence level of 95%.

### 2.3. Data Collection Procedures

The participants were recruited online by resorting to “snowball” sampling adapted to the virtual environment, which has been consolidated by other studies [40,41,42,43]. By means of this method, the participants themselves are responsible for recruiting other individuals in a similar situation through their social and contact networks. Following the method’s criteria, 30 MSM were initially selected, who were called “seeds”, with different characteristics, namely their region or district of residence, race/skin color (White/non-White), income and schooling level (higher education or not), so that these participants could indicate other MSM in their social networks to carry out this research.

The seeds were identified by means of two dating apps based on geolocalization (Grindr and Hornet), via direct chats with online users. To qualify as “seeds”, these participants had to meet the inclusion criteria of the study, be online at the moment of collection and agree to be contacted by email or telephone. The users included were the first active individuals in the platform in each of both apps who met the inclusion criteria, as recommended by previous studies [40,41,42,43].

Concurrently, the researchers also promoted the research on two social networks, Facebook and Instagram, targeting the MSM population over 18 years of age in both countries. Social networks were used as an additional resource due to their ability to access people located in inland, which is absolutely necessary in the case of a continental country such as Brazil. Only individuals who identified themselves as men (cisgender or transgender) and aged over 18 years old were included. Tourists and men who did not speak Portuguese were excluded. We also excluded those who did not complete more than 50% of the data collection form.

In total, 9112 MSM responded to the survey, 492 of which were excluded as detailed in Figure 1, with 8620 HIV-negative MSM being eligible to participate. Thus, 1682 of these MSM were using PrEP, with 970 (19.3%) of these residing in Brazil and 712 (21.5%) residing in Portugal (Figure 1).

### 2.4. Data Collection Instruments

The survey form was hosted on the SurveyMonkey platform for data collection in two versions, offering security features that allowed only one answer per internet protocol (IP). As there are important linguistic differences between both countries in the study, the form was made available in two versions: Brazilian and European Portuguese. The form was previously validated (face–content validation type) by 10 evaluators who specialized in the research topic (five from each country) regarding the objectivity, clarity and relevance, with a content validity index of 0.86. There was also a pre-test on five participants from each country before the survey was made available.

The form was divided into five sections with 40 questions, mostly of the multiple choice type. The questions addressed diverse social and demographic information (gender identity, sexual orientation, age, schooling, country of residence, country of origin, time living in the country), sexual and affective relationships (type of partners, type of relationship(s), number of partners), knowledge about ways to prevent HIV/AIDS, sexual behaviors and practices, protective measures adopted, seeking and using health services, and consumption and willingness to use PrEP.

### 2.5. Outcomes

The main outcome variable of this study was adherence to the use of PrEP (Y/N), which was examined with the following question: “Have you taken, as prescribed by the health professional, a daily PrEP tablet in the last 30 days?” For this definition, we followed the official recommendations regarding adherence: “the ideal use of antiretroviral drugs in the closest possible way to that prescribed by the health team, respecting the doses and times” [26,27,44].

To understand the factors associated with this use, social and demographic characteristics were investigated, as well as variables related to sexual and affective relationships, HIV testing and status, serophobia (fear of disclosing one’s HIV serological status), recent sexual behaviors and practices (last 30 and 60 days), persistence in condom use (defined as use in the most recent sexual intercourse) and reasons for not using condoms. The analyses were carried out by considering the population of the countries separately, dividing them according to country of residence and respecting the specificities of the countries in the analyses.

The following practices were defined:Chemsex: consumption of a drug immediately before and/or during sexual intercourse that is capable of altering the subjects’ perception and causing negligence in the use of protective measures against HIV [45]. In our study, we considered the following drugs alone or in combination: gamma-hydroxybutyric acid or “*Gisele*”, alkyl nitrites or “poppers”, and methamphetamine or mephedrone in the past 6 months.Fisting or footing: anal penetration using the fist or foot.Double penetration (DP): simultaneous sexual penetration by two or more penises.Cruising: free, consensual and anonymous sex practiced between men in public spaces, such as parks, bushes, beaches or parking lots.Challenging sexual practice: adherence to the recurrent use of two or three of these practices, defined according to the circumstances in which they take place.

### 2.6. Data Analysis

The data were evaluated using IBM SPSS Statistics 24.0 (SPSS Inc., Chicago, IL, USA). We adopted three levels of analysis, namely univariate analysis, bivariate analysis and multivariate analysis. In the first one, a descriptive analysis was used, which included the absolute and relative frequencies.

For a bivariate analysis of the variables of interest in relation to adherence to PrEP, we calculated the prevalence ratios (PRs) to evaluate the unadjusted associations between the main outcome (PrEP use) and the variables of social and demographic characteristics, and sexual behavior and practices, as well as their statistical significance, using Pearson’s Chi-square test. Considering *p* ≤ 0.05 as the minimum significance value (for both sides), 95% confidence intervals were also established.

All variables were first analyzed to assess whether there was multicollinearity or not, following the tolerance coefficients and VIF (Variance Inflation Factor) parameters. Considering the high frequency of the reference outcome (PrEP use > 10% or not), the association measure from traditional logistic regression analyses (odds ratio, OR) overestimated the associations. Thereby, we opted for the Poisson regression model with robust variance estimation using a covariance matrix (generalized linear model) to estimate the prevalence ratio (PR), which, in turn, is the most appropriate measure for cross-sectional studies. A logarithmic link function and the 95% CIs were also used.

For the multivariate analysis, the selection of the variables was made according to the results of the bivariate analysis, based on statistical significance (*p*-value ≤ 0.20), theoretical relevance or better adjustment conditions. The parameters observed for the best performance adopted the Akaike Information Criterion (AIC), log-likelihood, the omnibus test and effect tests (Type III) as references.

### 2.7. Ethical Considerations

The research was approved by the Research Ethics Committee of the IHMT belonging to Universidade Nova de Lisboa (Protocol No. 12.19/2020), as well as by the Research Ethics Committee of the University of São Paulo, Ribeirão Preto College of Nursing, Brazil

(Protocol No. 4163084). All the ethical norms in force in both countries were respected by applying the Informed Consent Form online to obtain the participants’ agreement. At the end of the research, they had access to institutional websites to obtain diverse information on HIV/AIDS prevention.

## 3. Results

The participants of this study were 1619 MSM, 970 living in Brazil and 712 in Portugal. It is noteworthy that the vast majority of these MSM (76.5% in Brazil and 78.2% in the case of Portugal) belonged to a metropolitan area, that is, a densely populated urban core; their characteristics are displayed in Table 1. The prevalence of PrEP use was 18.3% (*n* = 970) for Brazil and 21.5% (*n* = 712) for Portugal. 

In the bivariate analysis, 19 of the variables investigated were related to a higher prevalence of PrEP use, among which the following stand out. Having two or more partners in the last 30 days was associated with a 49-fold higher prevalence of PrEP use, while individuals who lived in a polyamorous relationship had a threefold higher prevalence of PrEP use (Table 1). Disclosing one’s serological status in mobile apps was associated with a fivefold higher frequency of PrEP use. The type of sexual practice adopted was also related to a higher prevalence of PrEP use, such as cruising (PR: 5.18), bareback sex (PR: 1.71), fisting or footing (PR: 1.47), double penetration (PR: 1.43) and having sex with three or more individuals (PR: 1.34).

In the multivariate analysis, we carried out modeling for the samples of residents in Portugal and Brazil, where two models were developed to assess how the variables behaved. In the Brazilian model, 19 variables were included, and in the model of those living in Portugal, 17 variables were included (Table 2). It is interesting to notice that some variables behaved differently according to the country. For example, whereas being an immigrant was associated with a higher prevalence of PrEP use in Portugal (PR: 1.36; *p*-value < 0.001), in Brazil, thr prevalence was reduced (PR: 0.83; *p*-value < 0.001). Whereas in Portugal, knowing the partner’s serological status increased the prevalence of PrEP use (PR: 1.28; *p*-value < 0.001), in Brazil, this effect was caused by not knowing the partner’s serological status (PR: 2.24; *p*-value < 0.001).

## 4. Discussion

In this study, we investigated the factors associated with PrEP use among HIV-negative MSM living in Brazil and Portugal, two countries that share a high and recurrent flow of people, highlighting opportunities and strategies of HIV prevention for the global health scenario. Our findings pointed to specificities regarding the differences and opportunities for the distribution of and access to PrEP use by MSM in the health services in their respective countries, which are added to the unique contexts of migratory flows, social determination and sexual practices themselves. However, the realities of MSM living in these countries and the very different social, cultural, economic and political contexts between them influence this population’s awareness of PrEP as well as their willingness to use it consistently, and may help explain the differences in the main findings.

Official data indicate that nearly 940,000 people in 83 countries received oral PrEP at least once in 2020. This represents a 49% increase in relation to the 630,000 PrEP users reported in 2019 and more than 2.5 times the number of PrEP users in 2018 (370,000). Most of the PrEP users in 2020 were reported in Africa (52%) and the Americas (30%) [7].

Our findings indicate that, although PrEP has been made available free of charge by the health systems of both countries since 2017 and the fact that MSM are a priority group in HIV prevention in both countries, fewer than one-quarter of the participants stated that they used this prophylactic measure, which is very close to the prevalence of use found in other recent surveys carried out in these countries [6,46]. 

The availability of PrEP in Europe, of which Portugal is a part, is fragmented, complex and in flux [39,47]. Among its member countries in 2016, only France reported that PrEP was nationally available and reimbursed, while data for the year 2019 [46] showed that 22 of the 55 countries indicated that reimbursed PrEP was available from their national health service, either through insurance or paid for by the public sector. In 2019, Portugal reported 718 PrEP users through its health system, while France reported 25,229 users and Cyprus only 10 [39]. Furthermore, in most of these countries, as in Portugal, PrEP is mainly provided in medicalized settings, with public infectious disease clinics being the most common setting for its provision [48], although research has indicated that this can create barriers to access for priority groups such as MSM [49].

In the case of Brazil, between January 2018 and December 2020, 32,791 individuals sought PrEP in the Brazilian public health services, with 29,467 (90%) receiving at least one dispensation (i.e., use for 30 days). However, only 16,938 users were on PrEP in Brazil at the end of December 2020 and 12,529 individuals (43% of those who started PrEP between January 2018 and December 2020) had discontinued its use, with no return, indicating significant discontinuity. Of the 16,938 PrEP users in December 2020 in the country, 82% (13,850) were gay and the others were cisgender MSM [30]. 

Within the Latin America and Caribbean region, there are also large disparities in the use and distribution of PrEP that reflect the heterogeneity of this region. In 2021, 22 countries reported having dispensed PrEP, with 57,902 people using it, of whom 48,230 people lived in Latin America and 9672 people lived in the Caribbean. Brazil led the way with over 40,000 users, while Paraguay had just 167, and Uruguay reported 38 [49].

In the case of both Brazil and Portugal, the national PrEP protocol suggests its use in people at a high risk of HIV infection, especially those who belong to key populations, those who have high partner turnover and those who tend to have sex without using condoms [6,23,26,27,30]. In our study, such practices increased the chances of adherence to PrEP, especially having a high number of recent partners, considering the results of both countries, or even considering the countries and their specificities separately. Challenging sexual practices, which comprise a set of important risk markers for HIV infection, were also determinants for greater PrEP use. According to these findings, MSM who engage in high-risk HIV practices seem to be aware of the need for prevention through PrEP, pointing to an important niche for public health interventions, either in the locations where these practices take place or in return PrEP consultations.

Aspects linked to disclosing one’s HIV serology or not knowing the partner’s HIV serology also influenced the adherence to PrEP. This finding is reinforced by the literature [49,50], which points out that having a steady partner, claiming/appearing not to have STIs and having been recently tested are associated with a certain credibility that leads to waiving condom use, which can lead the participants to seek additional prevention through PrEP [50,51,52].

Disclosing the HIV status via dating apps also increased the likelihood of adherence to PrEP use in both countries. This finding may be related to the fact that more than three-quarters of the study sample was concentrated in metropolitan areas, which corresponded to large population centers in both countries. There are studies that have indicated the role of dating apps for facilitating meeting people with common goals, and this finding may also mean that MSM in these countries have found this function of the apps to be an important means for mediating preventive practices, such as seeking out other PrEP users [40,41,42]. 

Our findings show an opportunity to improve the consolidation of PrEP from a global health perspective, especially when considering the migrant population, which is highly vulnerable. Whereas in Portugal, being an immigrant was associated with a higher prevalence of PrEP use, in Brazil, it reduced that probability. This may mean that the Portuguese health system is more effective than its Brazilian counterpart in identifying immigrants and incorporating them into the PrEP services. In fact, Portugal has exclusive community-based HIV testing and counseling services for MSM and immigrants, which may make a difference in the fight against the virus [51,53].

The fight against HIV/AIDS continues to face current challenges, such as disparity in access to the health system across different countries, even if they have agreements and similar sociocultural contexts, such as language. The legislation for the use of health systems by the immigrant population in Brazil (Sistema Único de Saúde) and Portugal (Sistema Nacional de Saúde), for example, reveals disparities, which have impacts on the possibility of using preventive technologies combined with PrEP. In this regard, there are data that indicate that health providers in some countries are afraid to invest in migrant populations, especially those who are undocumented, with regard to the provision of PrEP due to the possibility of a return to the country of origin soon, alongside discontinuity of service, time constraints, financial constraints and limited options [54].

In view of this, certain overlaps in health vulnerabilities may emerge as a result of the weaknesses of public policies in terms of their implementation and the reduced capacity for international relations. PrEP provision programs are regulated by and integrate the health systems of each country, in accordance with the most current and important clinical and public health guidelines, in their respective spheres of action (in Brazil, they are in the charge of the Ministry of Health; in Europe, they are the responsibility of the European AIDS Clinical Society [EACS]; and worldwide, they are the responsibility of the WHO) [51,52].

Since its implementation, PrEP use in Portugal has been recommended for key populations, although it was only from 2018 onwards that it became available exclusively through a reference hospital for HIV infection. For the users registered in the Portuguese National Health System (Sistema Nacional de Saúde, SNS), screening, monitoring, follow-up and provision of PrEP consultations are free of charge [51,52,53,54,55,56].

In turn, Brazil was the first Latin American country to offer PrEP free of charge via the SUS, initiating its distribution at the end of 2017 [28]. In this country, the medication is available from specific health services that are duly trained to offer prophylaxis, with the possibility of its prescription by physicians and nurses [57]. 

Although access to PrEP in both countries is free of charge, with similar dispensing plans, there is still a need to increase awareness and dissemination of PrEP use. With this strategy alone, potential users in the context of high HIV infection risk will be able to know this means of prevention and motivate themselves to initiate prophylaxis.

It is important to be aware of the inequalities in access to health services, which are reflected in the access to and knowledge about PrEP. In Portugal, regular testing, above the targets set by UNAIDS, increased the likelihood of PrEP use, reinforcing the importance of bringing these subjects closer to the health services and improving their knowledge of the various prevention means available [32,33]. However, the uptake of PrEP could be increased by exploring more opportunities for collaboration with community organizations to provide PrEP. If it insists on maintaining an exclusively hospital-based dispensation model, Portugal needs to increase the number of hospitals involved and its capacity to respond to demand in a timely manner, especially with a focus on the migrant population, which, for years, has been disproportionately affected by HIV/AIDS.

In the Brazilian case, in December 2021, the possibilities of PrEP care and follow-up were expanded with the possibility of prescriptions from private health services and withdrawal through public services, which has led to an expansion in the number of users [58]. However, PrEP is still very concentrated in large urban centers, and there is a need to review priority populations, such as people deprived of liberty and immigrants [30].

Despite all the social, political, cultural and economic differences, Brazil and Portugal have a similar HIV/AIDS prevention policy, especially regarding PrEP [23,25,27,29,30], with the exception of the important differences already mentioned in this text. However, there is no mechanism that facilitates the use of PrEP by people who migrate between these two countries, much less a specific flow for this purpose [23,25,26,27,30], thus creating an important programmatic vulnerability in view of the growing population in both countries that is doubly invisible with regard to HIV/AIDS.

At the global level, it is also necessary to be vigilant about the stigma related to PrEP use, especially by the most vulnerable groups (key populations), which may result in greater or lesser adherence to its use, especially when people reveal via dating apps that they are using PrEP. Seeking public health interventions that have proven to be successful in implementing and consolidating PrEP in various groups can promote better social acceptance [59].

We recommend expanding the global health actions for PrEP use locally and in the Ibero-American context investigated, considering the strategic position between the countries [55], the high number of MSM in immigration and the possibility of improving population coverage to prevent HIV transmission. Thus, it is relevant to consume diverse evidence of good practices and experiences already implemented by countries such as Kenya, Malawi and Zimbabwe, which may indicate effective possibilities for investing in health in this context [60].

Our research does have some important limitations. First, the observational design of the study, with the use of multivariate analysis, enabled the identification of possible causal relationships between the independent variables and adherence to PrEP use; however, this did not make it possible to establish the causal relationships between these factors. Secondly, the data were collected online as self-reported information, making it impossible to verify their veracity; in addition, the use of an electronic form and the need to read to answer the questions limited the sample to MSM with some level of educational instruction and with sufficient purchasing power to have access to a smartphone/computer and the, internet, which can be considered to be an important selection bias. Another limitation refers to the social, economic, cultural and political differences between the countries that have influenced their population’s willingness to “seek health” and the population’s own awareness of PrEP and its willingness to use it consistently.

It should also be considered the fact that the research was carried out during the COVID-19 pandemic and the related isolation measures, and thus closed or overcrowded health units may have affected the willingness/ability of participants to access PrEP, even decreasing the continuity of use of the same, although research has pointed to the growth of unprotected sexual activity in the context of COVID-19 in these countries [42,43,44]. In addition, the definition of “adherence to PreP” may vary in the literature due to the way this question was asked, and which time of use was considered. Finally, using “snowball” sampling was also a limitation, as this does not allow one to generalized the results to the general MSM population in these countries.

To overcome the listed limitations, we sought to establish generalization criteria by reducing biases and using the definition of “adherence” recommended by the official bodies of the countries.

## 5. Conclusions

Our findings indicate that, in our sample of participants, having more sexual partners and being fond of challenging and bareback sexual practices are the main factors that increase the likelihood of adherence to PrEP in Brazil and Portugal. In isolation, the partner’s serological status and being an immigrant were determinants that revealed different results when considering the Brazilian and Portuguese realities, as knowing the sex partner’s serological status and being an immigrant were associated with PrEP use in Portugal, unlike in Brazil, where they reduced adherence to PrEP.

Our results, respecting the specificities of each country, can provide support for the Brazilian and Portuguese health organizations, so that, based on knowledge of the social and demographic characteristics of this sizable sample of PrEP users, in addition to the sexual behaviors adopted, it will be possible to define strategies that encourage the adoption of preventive and care routines, especially when using this medication. On the other hand, we put emphasis on the participants that do not use PrEP, favoring knowledge about the public policies of the Brazilian and Portuguese governments regarding the target population and favoring transnational action, based on the principles of global health.

## Figures and Tables

**Figure 1 ijerph-20-04881-f001:**
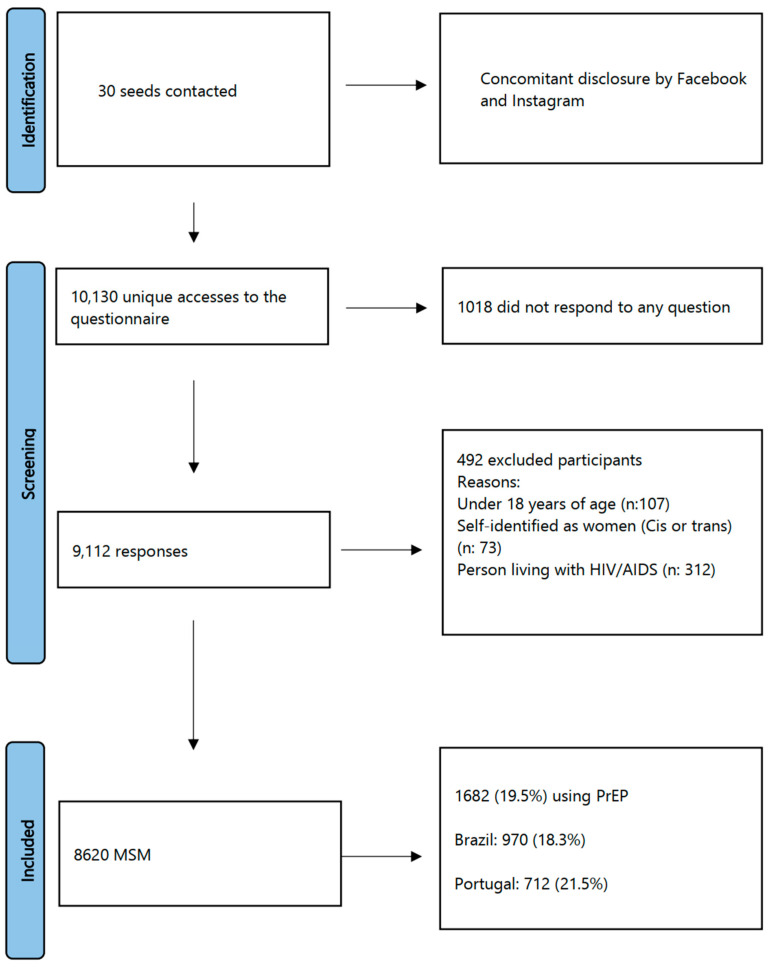
Flowchart of the selection and inclusion of the participants.

**Table 1 ijerph-20-04881-t001:** Sociodemographic characteristics, practices, sexual behaviors and bivariate analysis of the factors associated with the use of PrEP, 2020.

Variables		PrEP Use			
		Portugal (*n* = 712)		Brazil (*n* = 970)	
		*n* (%)	PR (95% CI) *p*-Value	*n* (%)	PR (95% CI) *p*-Value
**Social and demographic characteristics**					
Immigrant	Yes	387 (32.1)	2.0 (1.8–2.4) <0.001	171 (20.7)	1.2 (1.0–1.3) 0.046
	No ^[ref]^	325 (15.5)		799 (17.8)	
Age	<35 years old	517 (21.4)	1.0 (0.9–1.1) 0.817	746 (18.1)	0.96 (0.84–1.10) 0.570
	≥35 years old ^[ref]^	195 (21.8)		224 (18.8)	
Schooling level	Low	158 (18.4)	0.8 (0.7–0.9) 0.01	292 (21.9)	1.3 (1.1–1.5) <0.001
	Higher Education ^[ref]^	554 (22.6)		678 (17.1)	
Attracted to women	Yes	168 (24.0)	1.14 (0.98–1.33) 0.073	233 (22.8)	1.32 (1.16–1.51) <0.001
	No ^[ref]^	544 (20.9)		737 (17.2)	
Type of relationship	Single	501 (21.6)	1.32 (1.11–1.58) 0.002	734 (19.7)	2.00 (1.67–2.39) <0.001
	Polyamorous	81 (42.4)	2.60 (2.07–3.27) <0.001	108 (39.6)	4.03 (3.23–5.02) <0.001
	Steady partner/In a relationship ^[ref]^	130 (16.3)	-	128 (9.8)	-
**Sexual and affective relationships**					
HIV status	Known ^[ref]^	616 (21.6)	1.0 (0.81–1.18) 0.816	803 (17.3)	1.44 (1.24–1.66) <0.001
	Unknown	96 (21.1)		167 (24.9)	
HIV tests in the last 12 months	Yes	699 (32.2)	28.1 (16.3–48.4) <0.001	960 (26.6)	45.2 (24.3–84.2) <0.001
	No ^[ref]^	13 (1.1)		10 (0.6)	
Number of sex partners in the last 30 days	None ^[ref]^	3 (0.8)	-	3 (0.6)	-
	1	5 (0.5)	0.7 (0.1–2.7) <0.694 *	12 (0.6)	1.0 (0.31–3.86) <1.000 *
	≥2	704 (35.5)	44.2 (14.3–136.6) <0.001	955 (32.3)	54.49 (17.61–168.59) <0.001
Disclosing serological status in mobile apps	Yes	429 (53.4)	4.7 (4.1–5.4) <0.001	571 (50.3)	5.26 (4.71–5.87) <0.001
	No ^[ref]^	283 (11.3)		399 (9.6)	
**Recent sexual practices**					
Sexual intercourse with a person living with HIV	Yes	7 (10.8)	2.01 (0.99–4.07) 0.050	21 (18.8)	0.97 (0.65–1.43) 0.893
	No ^[ref]^	705 (21.7)		949 (18.3)	
Group sex and/or with 3 or more people	Yes	253 (18.8)	0.80 (0.70–0.92) 0.002	326 (14.7)	0.70 (0.62–0.80) <0.001
	No ^[ref]^	459 (23.4)		644 (20.8)	
Gouinage	Yes	4 (22.2)	1.03 (0.43–2.45) 0.942	6 (26.1)	1.43 (0.71–2.84) 0.209
	No ^[ref]^	708 (21.5)		964 (18.2)	
	No ^[ref]^	201 (15.9)		285 (12.6)	
Chemsex	Yes	290 (29.5)	1.62 (1.42–1.84) <0.001	368 (24.1)	1.52 (1.35–1.70) <0.001
	No ^[ref]^	422 (18.2)		602 (15.9)	
Fisting or footing	Yes	87 (32.1)	1.56 (1.29–1.88) <0.001	124 (25.1)	1.43 (1.21–1.68) <0.001
	No ^[ref]^	625 (20.6)		846 (17.6)	
Cruising	Yes	208 (79.1)	4.78 (4.32–5.28) <0.001	238 (79.9)	5.47 (5.00–5.97) <0.001
	No ^[ref]^	504 (16.5)		732 (14.6)	
Double penetration	Yes	223 (30.3)	1.60 (1.39–1.83) <0.001	268 (22.4)	1.31 (1.15–1.48) <0.001
	No ^[ref]^	489 (19.0)		702 (17.1)	
Challenging sexual practices	Yes	414 (32.1)	2.18 (1.91–2.48) <0.001	522 (25.3)	1.83 (1.64–2.06) <0.001
	No ^[ref]^	298 (14.7)		448 (13.8)	
Consistent condom use	Yes	65 (22.6)	1.05 (0.84–1.32) 0.623	106 (21.8)	1.21 (1.01–1.45) 0.033
	No ^[ref]^	647 (21.4)		864 (17.9)	
**Reasons for not using condoms**					
Interrupted practice or coitus	Yes ^[ref]^	172 (18.6)	1.21 (1.04–1.41) 0.013	234 (15.6)	1.23 (1.07–1.41) 0.002
	No	540 (22.6)		736 (19.3)	
Only insertive	Yes	32 (22.9)	1.06 (0.78–1.45) 0.691	40 (25.0)	1.38 (1.05–1.82) 0.020
	No ^[ref]^	680 (21.5)		930 (18.1)	
Partner reported PrEP use	Yes	105 (34.8)	1.72 (1.45–2.04) <0.001	151 (32.8)	1.94 (1.67–2.24) <0.001
	No ^[ref]^	607 (20.2)		819 (16.9)	
New/casual partner	Yes	540 (22.6)	1.21 (1.04–1.41) 0.013	736 (19.3)	1.23 (1.07–1.41) 0.002
	No ^[ref]^	172 (18.6)		234 (15.6)	
Partner stated they did not have an STI	Yes	66 (19.2)	0.88 (0.70–1.10) 0.285	103 (19.5)	1.08 (0.90–1.30) 0.420
	No ^[ref]^	646 (21.8)		867 (18.1)	
Partner reported a recent HIV test	Yes ^[ref]^	44 (15.7)	1.40 (1.06–1.85) 0.017	76 (16.0)	1.15 (0.93–1.43) 0.187
	No	668 (22.1)		894 (18.5)	

*p*-value (Pearson’s Chi-square test and * Fisher’s exact test); PrEP: pre-exposure Prophylaxis; PR: prevalence ratio; 95% CI: 95% confidence interval; STI: sexually transmitted infections, ^[ref]^: Reference.

**Table 2 ijerph-20-04881-t002:** Multivariate analysis of the factors associated with PrEP use among MSM from Brazil and Portugal, 2020.

Variables	β	aPR	95% CI		*p*-Value
			Lower	Upper	
**Portugal: multivariate analysis ^1^**					
≥2 casual sex partners per month	3.469	32.11	10.40	99.14	<0.001
Routinely undergoing HIV tests	2957	19.23	11.20	33.03	<0.001
Disclosing serological status in apps	0.741	2.02	1.88	2.34	<0.001
Being bisexual	0.393	1.48	1.33	1.64	<0.001
Being an immigrant	0.310	1.36	1.24	1.50	<0.001
Being fond of challenging sexual practices *	0.253	1.29	1.16	1.43	<0.001
Knowing the partner’s serological status	0.248	1.28	1.06	1.56	0.012
Frequent bareback sex	0.146	1.16	1.02	1.30	0.018
**Brazil: multivariate analysis ^2^**					
Routinely undergoing HIV tests	3.467	32.04	17.23	59.59	<0.001
≥2 casual sex partners per month	3.436	31.05	9.99	96.42	<0.001
Not knowing the partner’s serological status	0.806	2.24	2.03	2.47	<0.001
Disclosing serological status in apps	0.774	2.17	1.97	2.39	<0.001
Being in a polyamorous relationship	0.634	1.89	1.60	2.23	<0.001
Being single	0.355	1.43	1.25	1.63	<0.001
Being fond of casual sex	0.220	1.25	1.12	1.39	<0.001
Being fond of challenging sexual practices *	0.216	1.24	1.13	1.36	<0.001
Schooling level	0.126	1.14	1.03	1.26	0.014
Frequent bareback sex	0.120	1.13	1.01	1.25	0.027
Group sex	−0.168	0.85	0.76	0.94	0.002
Being an immigrant	−0.192	0.83	0.74	0.92	0.001

95% CI: 95% confidence interval. aPR: adjusted prevalence ratio; the best Akaike and likelihood criteria were evaluated. ^1^ Omnibus test (*p* < 0.001)/ROC curve: 0.933 (0.925–0.942); *p* < 0.001; AIC: 2377.4; LLR: −1178.7; deviance: 933.4. ^2^ Omnibus test (*p* < 0.001)/ROC curve: 0.938 (0.931–0.945); *p* < 0.001; AIC: 3395.8; LLR: −1683.9; deviance: 1427.8. * Cruising, chemsex, fisting or footing, and/or double penetration.

## Data Availability

Additional data are available upon request to the corresponding author.

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
