# Peer review of "Adherence to Pre-Exposure Prophylaxis (PrEP) among Men Who Have Sex with Men (MSM) in Portuguese-Speaking Countries"

_ijerph, 2023, doi:10.3390/ijerph20064881_

Round 1
Reviewer 1 Report (New Reviewer)
Comments and Observations on: “Adherence to Pre-Exposure Prophylaxis (PrEP) among Men who have Sex with Men (MSM) in Portuguese-speaking countries”
1. General: I found the work very interesting and the analyses were very meticulously done. However, there are few observations that the authors may need to considered.
2. Title: Abbreviation is not necessary in a title.
3. Abstract: The sub-title introduction in the abstract should be removed. Else, other subheadings such as objective, methods, results and conclusion may need to be added.
4. Why the choice of Brazil (South America) and Portugal (South Western Europe). Any scientific support? Are these countries dominant among the countries where there are higher users of PrEP? Could it be because they share historical, cultural and linguistic similarities (as claimed in line 65)? A few words to support this choice is necessary.
5. Keywords: the 16 words highlighted may seems to be too much for an article. Excessive Key words is akin to lack of focus of the article.
6. The placement of Portugal faraway from Brazil in the keywords could mean Portugal is just tangential to the main work. Keywords are arranged in their order of importance and not necessarily in alphabetic order.
7. Line 44, … and another 23 countries ..planned… and not plan
8. Line 53-56 highlighted the challenges inherent in PrEP use without any highlight on technological issue. However, the subsequent paragraph centers on strategies for solution which are to be implemented in countries with low technological incorporation. The two paragraphs are not properly linked. This would require reconciliation for proper flow of understanding.
9. Line 61-62: Is PrEP a national program or medicine taken to prevent getting HIV. This ought to have been highlighted earlier in the work before getting to line 67/68. Where the pre-exposure is a program, author should clearly indicate it with refs. The interchange of PrEP as a pill and as a program is confusing.
10. The shortcomings in the creation and implementation of PrEP in Brazil were amplified in the text. However, there was a silence on the challenges the program is facing in Portugal.
Methods
Population, sample
11. How did the author know when the sample is complete?
The expected sample size is not indicated. This would prevent the establishment of attrition rate to the questionnaire. Sample size determination techniques and the result is expected.
This will dictate when the sample selected is sufficient or not
12.
Data collection procedures
13. The restriction in educational attainment to (Elementary School, High School or undergraduate studies), is not known. Perhaps, exclusion of individuals without education could be obvious due to platforms used (social media). But what happened to graduates? Why were they excluded.
14. Line 137. The following ‘are’ defined.
15. Line 148 - Statistical Package for the Social Sciences (SPSS) and not SPPS,
Line 148- This version of SPSS supposed to be referred to as IBM SPSS Statistics 24. Authors to reconfirm.
Data analysis
16. I was expecting the authors to highlight first that: three levels of analysis were adopted, namely: univariate analysis, bivariate and multivariate analysis and what was done under each of the level of analysis. This would have made the methodology clearer for reproducibility by other researchers.
17. The procedures as presented here were muddled up.
18. Outcomes line 129
Query: Outcome variables or outcome?
These are not the same things.
Outcome variables are Dependent variables (measured variables)
Outcome is a result (maybe, the result from data analysed).
19. Line 149-151: The steps on bivariate analysis seems like incomplete statements
For example, what are the variables involved in the computations of the associations?
Prevalence Ratios (PRs) were used to evaluate unadjusted associations (bivariate analysis) between or among which variables.
20. This section is supposed to highlight the variables involved under the bivariate analysis and the dependent and independent variables for the multivariate section. The sudden jump to the report that the outcome variable (PrEp) has high frequency is worrisome.
21. Line 157-159 - /Traditional logistic regression vs Poisson regression model
Was there a compelling rule that traditional logistic regression analysis must be used for this type of study? Hence the picking and dropping the technique makes no sense. The analysis plan supposed to have indicated the appropriate statistics to use rather a system of trial and error.
22. Ethical Considerations (Line 167)
Anything to validate the claim that the institutions mentioned endorse the project instrument would have been better. Any ethic certificate number?
Results
23. If the sample is 8260, what then account for different n’s (n = 6,535; n = 6,425 & n = 6,054).
Attrition rate could not be established from the beginning.
24. Table presentation
The first table has no title. However, it was referred to as Table 1.
Also, it is expected that most profound results should carry table reference. As it is, the readers are left with the task of looking for the table that carries the actual result. Reference to Table in the text are too scanty/meagre.
25. Also, using a single block of Table to present a number of results (focus areas) seems not good.
I suggest the use of separate Table to present the information here such as:
Table 1. Social and demographic characteristics
Table 2. Sexual and affective relationships
Table 3. Recent sexual practices
etc, etc.
26. The calculation here is not clear in summation along row or column
Discussion
27. Query:
The discussion is based on a study conducted among HIV-negative MSM in with two countries (Brazil and Portugal) whereas, the introduction (line 76-80) and the methodology (lines 88-92, 97-102, 107-108, etc) do not suggest that the sample are HIV-negative MSM at all. There is therefore a great disconnect.
Line 175 is specifically that ……the participants of this study were 8,620 MSM….
The statement is seemed contradictory to the sample drawn and what the objective earlier stated.
28. I could see new literature/citations in Discussion Section.
This is an element of Post-Study (post-result) literature review.
The Discussion Section is only meant for validation of issues earlier raised with the results from the data analysis. New literature are not expected in the discussion section. The author discusses the key findings whether they in tandem or contradiction with existing studies reviewed. Item/Ref 29, 30, 31-34 (up to 42) seems to be new literature (as it were). The practice of post-study literature review is not strong in academic/research parlances.
29. Recommendation (Lines 332-337)
Do you make recommend at the Discussion Section?
Recommendations are expected after the conclusion
Most studies write: Conclusion and recommendation and not recommendation and conclusion.
Thus, I suggest the same sub-heading for the authors- Conclusion and recommendation.
The two paragraphs in the conclusion should be merged to one.
Apart, from Brazil and Portugal, what suggestion could be made for other countries and world at large based on the key findings from the study.
30. Limitations of the study
While this section or paragraph is meant to present the specific limitations inherent in this study, the author may also need to highlight what mitigating steps took to address these challenges.
Finally, this is very tangential
31. The Author Names: Consistency in names presentation is crucial in scientific community. The first author for instance has ÁFL Sousa, FL Sousa, AF Lopes de Sousa for different articles. I suspect the author will have different names in Scopus and Google Scholar.
Author Response
Please see attachment.

Reviewer 2 Report (New Reviewer)
This paper has the potential to contribute to the current literature regarding PrEP access, awareness, and Willingness to Use.
The major point that I can’t get through is how the authors defined the main outcome of the study - “adherence” or “willingness to use PrEP consistently” or “likelihood of adherence to PrEP”. Without having this definition clear, no conclusions can be drawn about the answers for the respondents.
Throughout the manuscript, the authors refer to adherence to PrEP. In my understanding, adherence refers to the extent to which patients take medication as prescribed by their doctors. It is not clear if the respondents were asked if they had or not a prescription.
The main outcome of the study was accessed by the question: “Have you taken a daily PrEP tablet in the last 30 days?” If the respondents have a prescription, the answer “N” may refer to low adherence. If the respondent does not have a prescription, the answer “N” does not give a clear conclusion (Was it low access to PrEP? Was it low awareness of PrEP? Was it low willingness to use PrEP?).
Brazil has extensive literature on PrEP Awareness, knowledge, willingness to use, and adherence that may help the authors to clarify these definitions.
Please address the comments to add more clarity to some of the interpretations made.
-
In the Introduction, lines 108 to 110, The authors state that: “It is in this international scenario that this study aimed at analyzing the factors associated with adherence to PrEP among MSM from two Portuguese-speaking countries (Brazil and Portugal), highlighting the differences, similarities, opportunities and preventive strategies for the global health scenario.”
In Methods, Population, sample, and eligibility criteria, lines 118 to 123, the prescription of PrEP is not an inclusion criteria.
In Methods, Data collection instruments, lines 161 to 166, the authors states that “The questions addressed (...) and willingness to use PrEP”.
And finally, In Methods, Outcomes, lines 168-169, the authors states that “The main outcome of this study was PrEP use (Y/N)”.
I may not be understanding this correctly, but for me, it is unclear how the authors define “adherence”; the difference between adherence, "willingness to use PrEP” and “PrEP use in the last 30 days” for this study; and how the adherence data were collected in the survey (Was it only collected for the group that used PrEP in the last 30 days?). Please clarify.
-
Materials and Methods, line 176: Please revise the english (some words are written in “portuguese”)
-
Results, lines 222 to 223: Please review the text. It seems that the sentence is incomplete, “truncated”
-
Results, line 241: Please revise the english (some words are written in “portuguese”)
-
Results, lines 241 to 246: In my opinion, this clarification could be presented in the "Discussion" section.
-
Discussion, lines 301-302 and throughlong the Discussion section: Please, I may not be understanding this correctly, but I believe there may be a typo (or perhaps some confusion) regarding the definition of “adherence” and “willingness to use PrEP consistently”. Please clarify and/or define adherence on “Methods”.
-
DIscussion, lines 375 to 381: Please please reference this paragraph. If this affirmation is based on the author's expertise, please clarify it in the paragraph.
-
In my opinion, the Discussion section is too long. This section aims to discuss your findings (rather than those of the other researchers) and each paragraph should explore your results. The previous studies are presented in this section to explain, reinforce, and refute your findings or bring critical evaluations, and learning points about your research and only studies related to your results should be discussed.
-
Discussion, lines 272 to 276: Here the authors state that the prevalence of the MSM using PrEP in the study is below the use prevalence found in other recent surveys and cited the “The EMIS Network”. The authors did not take into account that the questions used in both surveys are different and the answers will vary depending on how the question is formulated. While in the present study, the respondent is asked if “Have you taken a daily PrEP tablet in the last 30 days?”, in the cited reference the respondent is asked if “Have you ever taken PrEP?”.
In spite of that, please review the reference, I am not sure that the authors state a prevalence higher than ¼ in the cited article.
Author Response
Please see attachment

This manuscript is a resubmission of an earlier submission. The following is a list of the peer review reports and author responses from that submission.
Round 1
Reviewer 1 Report
1. English language editing required.
2. The literature presented in the discussion should rather be presented in the introduction. A better understanding of the MSM situation in Brazil and Portugal would help the readers better understand the reason for doing the study.
3. Given the snowball sampling methodology, inference to the population cannot be made. Thus the entire discussion, limitations, and conclusions/recommendations need to change.
Author Response
Dear reviewer, thank you very much for your excellent comments. We take into account and implement all suggestions. We detail them below:
1. English language editing required.
A1 - We did a deep and detailed review of the entire language with trained proofissional.
2. The literature presented in the discussion should rather be presented in the introduction. A better understanding of the MSM situation in Brazil and Portugal would help readers better understand the reason for doing the study.
A2-We tried to present all the interesting literature, but the comment did not seem accurate. The manuscript itself is huge, and we wouldn't want to make it any bigger. We believe that is the problem of course, but we clarify further:
Brazil and Portugal have similarities that deserve to be studied, especially when one considers that for centuries the two countries have been going through a process of immigration of their population from one to the other, and vice versa.
3. Given the snowball sampling methodology, inference to the population cannot be made. Thus the entire discussion, limitations, and conclusions/recommendations need to change.
-We agree with the reviewer, and apologize if we imply otherwise. We modified the text, emphasizing that our data should be viewed with our sample in mind. However, it is a robust, well-diversified sample that brings important results for public policy.
Reviewer 2 Report
In the manuscript entitled “Adherence to Pre-Exposure Prophylaxis (PrEP) among Men who 2 have Sex with Men (MSM) in Portuguese-speaking countries” by Alvaro Francisco Lopes de Sousa et al., investigation of the effect of different parameters including the immigrtion status and serological status of the sexual partner on the use of pre-exposure prophylaxis in two Portuguese speaking countries Portugal and Brazil was performed. The study was carried out in accordance with the proposal of the project and the manuscript is written in standard English language. However, it will be better if the authors cite more relevant studies on this subject as given below.
Qu D, Zhong X, Xiao G, Dai J, Liang H, Huang A. Adherence to pre-exposure prophylaxis among men who have sex with men: A prospective cohort study. Int J Infect Dis. 2018 Oct;75:52-59.
Mannheimer S, Hirsch-Moverman Y, Franks J, Loquere A, Hughes JP, Li M, Amico KR, Grant RM. Factors Associated With Sex-Related Pre-exposure Prophylaxis Adherence Among Men Who Have Sex With Men in New York City in HPTN 067. J Acquir Immune Defic Syndr. 2019 Apr 15;80(5):551-558.
Author Response
Dear reviewer, thank you very much for your excellent comments. We take into account and implement all suggestions. We detail them below:
1. ...it will be better if the authors cite more relevant studies on this subject as given below.
Qu D, Zhong X, Xiao G, Dai J, Liang H, Huang A. Adherence to pre-exposure prophylaxis among men who have sex with men: A prospective cohort study. Int J Infect Dis. 2018 Oct;75:52-59.
Mannheimer S, Hirsch-Moverman Y, Franks J, Loquere A, Hughes JP, Li M, Amico KR, Grant RM. Factors Associated With Sex-Related Pre-exposure Prophylaxis Adherence Among Men Who Have Sex With Men in New York City in HPTN 067. J Acquir Immune Defic Syndr. 2019 Apr 15;80(5):551-558.
A1 - Thank you very much for your suggestion. We promptly added as per suggestion
Reviewer 3 Report
Dear authors.
The whole manuscript is well written. However, I have some suggestions to be considered by the authors and editor. Please send the manuscript for proofreading, typos, and grammar check.
Abstract:
- Ln 21. Please delete the introduction words.
- Please mentioned the gaps or problems.
- Would be informative if mentioned the study design.
- Please mentioned the month of an online survey.
- Ln. 33: hey population? What?
Introduction:
- The introduction is presented in many paragraphs. Please concise into crucial information.
- Ln 57-64: please provide some references to support the information.
- Please mentioned the variables or indicators that will be included in this study and what the theoretical framework used?
- The novelty statement is weak. Please elaborate by comparing it with the previous studies.
- Please put high concern in the study purpose. Ln 79: “highlighting the opportunities….. scenario”. I did not see this in the result or study findings. Please clarify.
Methods:
- What do you mean by international study in line 83? Should you mention it? I think we all know that this study is conducted at the International level since used two countries.
- In the sub-heading of population, sample, and eligibility criteria. Please provide detailed information about the sample/participant. How many are included? How many dropped out? How many groups? Exclusion criteria? Etc. Please provide the sample flow diagram for clear information.
- How if the participants do not complete the online survey? Or any missing answers from the survey?
- What are the criteria for “seeds” (ln 98)?
- It would be informative if the authors also mentioned the total number of participants that participated through each social media.
- Data collection instruments. Please mentioned the value of reliability and validity results. Do not only mention such presented in line 119-121.
- If the questions consist of categorical data, please mentioned it. And provide the reference of the questionnaire.
- In the statistical analysis, please mention one or two-sided.
Results:
- In the result section, the authors’ mentioned “general group”. What is that? No information in the method section.
- Author’s mentioned the subgroup analysis (Ln: 194). Please present the figure for subgroup analysis to know the wide range of each relationship.
- Why don’t you analyze the demographic information in the multivariate analysis?
Discussion:
- Please do not mention any statistical results or value in the discussion section. Please define it and discuss it.
- Discussion is too long. Please concise the specific findings.
- Please elaborate on the bias in this study in the limitation section.
Author Response
Dear reviewer, thank you very much for your excellent comments. We take into account and implement all suggestions. We detail them below:
Dear authors.
The whole manuscript is well written. However, I have some suggestions to be considered by the authors and editor. Please send the manuscript for proofreading, typos, and grammar check.
-Answer: Thank you very much. The manuscript was sent to a native English-speaking reviewer.
1. Abstract:
- Ln 21. Please delete the introduction words.
Answer: Fixed.
- Please mention the gaps or problems.
Answer: As suggested
- Would be informative if mentioned the study design.
Answer: As suggested
- Please mention the month of an online survey.
Answer: As suggested
Ln. 33: hey population? What?
Answer: It was a typo. we fixed
2.Introduction:
- The introduction is presented in many paragraphs. Please concise into crucial information.
-Answer: This was difficult to reconcile, considering that the previous reviewer asked to increase the introduction and the second to add more texts to it. We seek to be parsimonious, to serve all reviewers.
- Ln 57-64: please provide some references to support the information.
-Answer: We added as suggested.
- Please mentioned the variables or indicators that will be included in this study and what the theoretical framework used?
-Answer: This is detailed in the methods section of the study. We detail which variables we analyze, the background of the literature and how we work on the text.
- The novelty statement is weak. Please elaborate by comparing it with the previous studies.
-Answer: There are no previous studies that approach this object in this way and this is the great novelty of the study. We make this clearer in the introduction.
- Please put high concern in the study purpose. Ln 79: “highlighting the opportunities….. scenario”. I did not see this in the result or study findings. Please clarify.
-Answer: Thank you very much for this tip, we reinforced it in the introduction and in our suggestions in the results.
Method:
- What do you mean by international study in line 83? Should you mention it? I think we all know that this study is conducted at the International level since used two countries.
-Answer: In order not to cause problems of understanding, we removed the term.
- In the sub-heading of population, sample, and eligibility criteria. Please provide detailed information about the sample/participant. How many are included? How many dropped out? How many groups? Exclusion criteria? Etc. Please provide the sample flow diagram for clear information.
-Answer: We have added this information and the flowchart;
- How if the participants do not complete the online survey? Or any missing answers from the survey?
-Answer: In this case, the participant does not fit the survey. It was detailed that those who did not fill in the necessary questions were excluded.
- What are the criteria for “seeds” (ln 98)?
-Answer: We detail this information. The seeds were identified by means of two dating apps based on geolocation (Grindr and Hornet), via direct chats with online users. To qualify as “seeds”, these participants had to meet the inclusion criteria of the study, be online at the collection moment and agree to be contacted by email or telephone. The users included were the first active individuals on the platform in each of both apps and who met the inclusion criteria, as recommended by previous studies
- It would be informative if the authors also mentioned the total number of participants that participated through each social media.
-Answer: Unfortunately, the platform did not give us this information.
- Data collection instruments. Please mention the value of reliability and validity results. Do not only mention such presented in line 119-121.
-Answer: We used a questionnaire created by the authors themselves based on previous publications in peer-reviewed journals. The questionnaire was face-content validated by PhD researchers with expertise in the subject using the Delphi technique, an efficient and consolidated methodology to generate consensus based on the opinion of professional experts in the subject. The questionnaire was made available to the group of researchers online and evaluated regarding the degree of importance of each question for the research object, taking into account objectivity, clarity and relevance. For this, a Likert-type scale was used (1 – very small, 2 – small, 3 – reasonable, 4 – large and 5 – very large). Two rounds were held until consensus was reached. The content validity coefficient (CVC) was used to analyze the agreement index, so that, to remain on the form, the question needed to reach a minimum percentage of agreement, percentage fulfilled by all items. Subsequently, the questionnaire was tested (pre-test) with 10 participants from the reference population, with no need to make changes.
References:
Sousa AFL, Hermann PRS, Fronteira I, Andrade D. Monitoring of postoperative complications in the home environment. Rev RENE. 2020;21:e43161. DOI: http://dx.doi.org/https://doi.org/10.15253/2175-6783.20202143161
Hernandez-Nieto RA. Contributions to statistical analysis. Merida: University of Los Andes; 2002
- If the questions consist of categorical data, please mention it. And provide the reference of the questionnaire.
-Answer: All categorical questions included in the analysis have been entered.
- In the statistical analysis, please mention one or two-sided.
-Answer: We use statistically relevant on both sides. This was added.Results:
- In the result section, the authors’ mentioned “general group”. What is that? No information in the method section.
-Response: Thank you very much for this great comment. This was duly explained in the method and results in this new version;- Author’s mentioned the subgroup analysis (Ln: 194). Please present the figure for subgroup analysis to know the wide range of each relationship.
-Answer: This is explained at the beginning of the results:
"The participants of this study were 8,620 MSM, with their characteristics displayed in Table 1. The general prevalence of PrEP use, that is, considering both countries together, corresponded to 19.5% (n=1,682) of the overall sample: 18.3% ( n=970) for Brazil and 21.5% (n=712) for Portugal.The participants were predominantly young adults (75.8%; n=6,535), with high schooling levels (74.54%; n=6,425) and single (70.2%; n=6,054). It is noted that, among the PrEP users, 33.2% of the sample were immigrants".- Why don't you analyze the demographic information in the multivariate analysis?
-Answer: We included these variables in the bivariate analysis, however they were not significant to enter the multivariate analysis. That is, these variables did not influence the outcome to the point of being considered as fulfilling the statistical criteria.
Discussion:
- Please do not mention any statistical results or value in the discussion section. Please define it and discuss it.
-Answer: We have removed all references to numeric values. Thank you very much- Discussion is too long. Please concise the specific findings.
-Answer: It was very difficult to meet this criterion... see that we have 3 analysis models, a population of two countries considered together and analyzes of the same separately. There are so many details that we had difficulties not to make the discussion bigger. However, we have made additional effort and shortened the discussion as suggested.
- Please elaborate on the bias in this study in the limitation section.
-Answer: Thank you very much. We added all the biases we saw.
Round 2
Reviewer 3 Report
All comments are addressed.